# Chitosan–Surfactant Composite Nanocoatings on Glass and Zinc Surfaces Prepared from Aqueous Solutions

**DOI:** 10.3390/molecules29133111

**Published:** 2024-06-29

**Authors:** Péter Márton, Liza Áder, Dávid Miklós Kemény, Adél Rácz, Dorina Kovács, Norbert Nagy, Gabriella Stefánia Szabó, Zoltán Hórvölgyi

**Affiliations:** 1Centre for Colloid Chemistry, Department of Physical Chemistry and Materials Science, Faculty of Chemical Technology and Biotechnology, Budapest University of Technology and Economics, Műegyetem rkp. 3, H-1111 Budapest, Hungary; marton.peter@vbk.bme.hu (P.M.); aderlizalidia@edu.bme.hu (L.Á.); 2Department of Materials Science and Engineering, Faculty of Mechanical Engineering, Budapest University of Technology and Economics, Műegyetem rkp. 3, H-1111 Budapest, Hungary; kemeny.david@gpk.bme.hu (D.M.K.); kovacs.dorina@gpk.bme.hu (D.K.); 3Centre for Energy Research, Institute for Technical Physics and Materials Science, Konkoly Thege Miklós út 29-33, H-1121 Budapest, Hungary; racz.adel@ek.hun-ren.hu (A.R.); nagy.norbert@ek.hun-ren.hu (N.N.); 4Department of Chemistry and Chemical Engineering of Hungarian Line of Study, Universitatea Babes-Bolyai, 11 Arany Janos str., RO-400028 Cluj-Napoca, Romania

**Keywords:** chitosan, anionic surfactants, composite nanocoatings, hydrophobicity, hygroscopicity, corrosion protection

## Abstract

Hydrophobic coatings from chitosan–surfactant composites (ca. 400 nm thick by UV-Vis spectroscopy) for possible corrosion protection were developed on glass and zinc substrates. The surfactants (sodium dodecyl sulfate, SDS or sodium dodecylbenzenesulfonate, and SDBS) were added to the chitosan by two methods: mixing the surfactants with the aqueous chitosan solutions before film deposition or impregnating the deposited chitosan films with surfactants from their aqueous solutions. For the mixed coatings, it was found that the lower surface tension of solutions (40–45 mN/m) corresponded to more hydrophobic (80–90°) coatings in every case. The hydrophobicity of the impregnated coatings was especially significant (88° for SDS and 100° for SDBS). Atomic force microscopy studies revealed a slight increase in roughness (max 1.005) for the most hydrophobic coatings. The accumulation of surfactants in the layer was only significant (0.8–1.0 sulfur atomic %) in the impregnated samples according to X-ray photoelectron spectroscopy. Polarization and electron impedance spectroscopy tests confirmed better barrier properties for these samples (40–50% pseudo-porosity instead of 94%). The degree of swelling in a water vapor atmosphere was significantly lower in the case of the impregnated coatings (ca. 25%) than that of the native ones (ca. 75%), measured by spectroscopic ellipsometry. Accordingly, good barrier layer properties require advantageous bulk properties in addition to surface hydrophobicity.

## 1. Introduction

Chitosan (CS) is an environmentally friendly cationic biopolymer [1,2,3,4], widely used in the food industry [5], cosmetics [6], and even in corrosion protection [7,8]. Corrosion is a significant issue [9], and the utilization of eco-friendly, barrier-type coatings from biopolymers can provide real alternatives to conventional anti-corrosive coatings [10]. CS is made from the partial deacetylation of chitin by boiling it in a concentrated alkaline solution [11] or by biological extraction [12].

The surface of coatings made from water-soluble CS by dipping or casting is not hydrophobic, and the hygroscopic property of CS is also considerable [13]. In coatings of CS modified with non-polar side chains, the accumulation of the side chains on the surface can improve the barrier effect due to enhanced hydrophobicity [14]. The non-polar CS molecules, however, are no longer soluble in water; hence, only the use of organic substances that are more harmful to the environment would be suitable for film deposition.

Previously, it was verified that the water repellency and permeability of CS layers can be controlled by a subsequent acylation of the deposited native CS coating carried out in a methanolic solution [15]. However, it remains a challenge to achieve the beneficial (hydrophobic) properties of the coatings quasi-simultaneously with the coating deposition, and without the necessity of an additional treatment step. It would be advantageous to dissolve anionic surfactant molecules in the aqueous CS solution that can be attached to the positively charged CS molecules by electrostatic attraction and are sufficiently surface-active to adsorb in a certain orientation at the water–air interface. The surfactants would be trapped at the water-air surface, which would result in a more hydrophobic coating after drying; thus, a coating with favorable properties would be produced in one step. However, since the interfacial processes are significantly influenced by the interactions and structure formation of surfactants and macromolecules in the solution, they need to be investigated further.

These polyelectrolyte-surfactant interactions in aqueous phases have been extensively studied [16,17,18]. The interaction between oppositely charged polyelectrolytes and surfactants is complex and occurs even at a low surfactant concentration [19]. Amphipathic molecules bind to the polyelectrolyte with opposite charges, forming micelle-like aggregates, where the polyelectrolyte molecules couple to the surface of the micelles [20,21] (this can be considered a cooperative interaction between the oppositely charged micelle and the macromolecule). As the surfactant concentration increases, a two-phase system (precipitate) is formed [22]. Studies on the wettability of films formed by the drying of these complex aqueous solutions are practically absent from the literature.

The aim of this work is to develop hydrophobic chitosan-based protective coatings by exploiting the self-assembling properties of surfactants and to gain a deeper understanding of the factors influencing the barrier properties of such coatings by studying polymer–surfactant interactions. Two commercially available anionic surfactants, i.e., sodium dodecyl sulfate (SDS) and sodium dodecylbenzenesulfonate (SDBS) [23], were selected for the experiments. The surface tension of the surfactant-containing precursor solutions and the wettability of the deposited coatings, as well as the accumulation of the surfactants within the coating (by X-ray photoelectron spectroscopy, i.e., XPS), were studied. The barrier properties of the coatings were characterized in aqueous and humid environments by electrochemical and spectroscopic ellipsometry measurements. For comparison, surfactant-containing coatings were also made in two steps by impregnating the native CS coatings in aqueous surfactant solution.

The prepared coatings can be used in a wide range of applications, for example, temporary corrosion protection of metals in maritime transport (against humid and salty air) or to protect electronic circuits in a humid environment.

## 2. Results and Discussion

### 2.1. Surface Tension Measurements

In order to characterize the polymer–surfactant interactions occurring in the aqueous solutions, the surface tension of pure and CS-containing solutions of SDS and SDBS was determined by the pendant drop method for different surfactant concentrations. The measured values are summarized in Figure 1. From the data of the pure solutions, the critical micelle concentration (cmc) of SDS and SDBS was determined. For SDS, the obtained results are in good agreement with the data in the literature (8.02 mM [24]), and for SDBS, the value is in the range of those found in the literature (1.62 mM [25,26]).

Figure 1 shows that CS by itself slightly reduces the surface tension of solutions compared to pure water (from 72 mN/m to 69 mN/m) due to its weak surface activity. Surfactants in the presence of 1% CS behave differently. In the case of SDS, a rapid decrease can be observed between 0 and 0.2 mM, while in the case of SDBS, the value of the surface tension decreases only slightly at low concentrations, followed by a stronger decrease above 0.2 mM. This can be explained by the difference in the polyelectrolyte-surfactant interactions and the surface activity of the formed adducts, which can be influenced by the difference in the molecular structure of the two surfactants [27,28].

A minimum can be observed for both curves (at a concentration of 0.75 mM for SDS and 0.5 mM for SDBS). This is often associated in the literature with the critical association concentration (cac), which indicates the formation of strong micelle-like (cooperative) polymer–surfactant complexes (below cac, the non-cooperative binding is favored) [29]. The upper limit of the measurement in both cases was the aggregation of these complexes, at which point visible precipitation was observed (1.5 mM for SDS and 0.75 mM for SDBS). This (as well as the cac value) can be observed at lower concentrations in the case of SDBS, which can be explained by the stronger non-polar characteristic of the surfactant (also, see the smaller cmc value) [30].

### 2.2. UV-Vis Spectroscopy: Layer Thickness and Refractive Index

Nanolayers were formed from pure and surfactant-containing CS solutions using the dip-coating technique. Additionally, pure CS layers were impregnated in more concentrated surfactant solutions (30 mM SDS and 1.5 mM SDBS) for 30 min as a different modification technique (these values were determined based on previous preliminary experiments, which are not detailed here). To study the effect of the surfactant content on the properties of the layers, the thickness and refractive index were measured by UV-Visible spectroscopy, fitting a thin-layer optical model to the transmittance spectra (Hild model, see Reference [31]). The spectra for the studied concentrations are summarized in Figure A1 in Appendix A, while the fitted results are shown in Table 1. In the last column, the changes in layer thickness are summarized with respect to the thickness of the native CS layer (395 ± 7 nm). In the case of higher surfactant concentrations, the significant light scattering of the coatings did not allow the evaluation with the Hild model (due to the light scattering, the transmittance at shorter wavelengths decreased significantly; the first sign of this can be seen in the “Gls/CS+SDS (0.5 mM)” sample spectrum in Figure A1a).

The Landau-Levich equation can be used to interpret the trends in the layer thickness values, which estimates the thickness of the liquid film formed on the substrate during dip-coating (*d_l_*) as a function of the parameters of the coating deposition (Equation (1)) [32].
(1)$$d_{l}=0.94\cdot \frac{{\left(u\eta \right)}^{\frac{2}{3}}}{\gamma^{\frac{1}{6}}{\cdot \left(\varrho g\right)}^{\frac{1}{2}}}$$
where *u* is the withdrawal speed, *η*, *ρ*, and *γ* are the dynamic viscosity, density, and surface tension of the precursor solution, respectively, and *g* is the gravitational acceleration. As can be seen, a decrease in the viscosity of the precursor solution reduces the thickness of the settling liquid film (and thus the nanocoating), but a decrease in the surface tension has the opposite effect. For the mixed systems, the addition of the surfactant reduced the viscosity of the precursor solutions in both cases (see Figure A2 in Appendix A). Barreiro-Iglesias et al. observed a similar decrease in the viscosity of CS-SDS solutions, which became much stronger after reaching the cac of the system [33]. At the same time, the surface tension also decreased for both systems (but to a different extent, see Figure 1); thus, the result of the two effects caused the change in layer thickness, as shown in Table 1. The ca. 25% increase in layer thickness in the case of impregnated systems can be explained by the increase in volume due to the accumulation of surfactant molecules in the coating during impregnation.

### 2.3. Wettability: Contact Angle Measurements

To quantify the wettability of the samples, advancing and receding values of water contact angles were measured for the mixed and impregnated systems (Figure 2). The hydrophobicity of the surfaces increased in all cases because, during the deposition, the surfactants (preserving their oriented adsorption at the water–air interface) accumulate at the surface of the coatings.

In the case of mixed systems (solid markers in Figure 2), the advancing and receding contact angles increase at low surfactant concentrations (by ca. 20° for SDS and 10° for SDBS). This is due to the increase in the interfacial amount of the surface-active molecules. However, when the surfactant concentration increases further, the surface once again becomes more polar (the advancing contact angles decrease to 80° and 70°). For SDS, this occurs near the cac (0.5–0.7 mM, see Figure 1a), while for SDBS, this occurs at the beginning of the sharply decreasing section of the surface tension–concentration curve (0.1–0.2 mM, see Figure 1b), which suggests that the phenomenon correlates well with the polymer–surfactant interactions in the aqueous phase.

For the coatings impregnated with SDS (open markers in Figure 2a), the contact angles are roughly the same (85–90°) as in the case of the most hydrophobic mixed system (0.5 mM). The reasons for this may be that, in both cases, the surface of the coatings is fully covered with SDS molecules and this hydrophobic surface layer determines the contact angle value. In the case of SDBS, on the other hand, there is a significant difference between the contact angle values of the most hydrophobic mixed system (0.1 mM) and the impregnated system (Figure 2b). If the maximum amount of surfactant accumulates on the surface during impregnation (resulting in an advancing water contact angle of over 100°), then it follows that there is significantly less SDBS on the surface in the case of the mixed system. This may originate from the difference in the interactions: in the case of SDBS, the association in the bulk phase may be more favorable than the surface accumulation. Hence, the smaller amount of the surfactant (or the presence of the less surface-active adducts) cannot render the coating as hydrophobic as the impregnated sample.

Since the goal was to create coatings with a hydrophobic surface and good barrier properties, the films with the highest advancing water contact angles were selected for further tests (along with the native CS coating). The long-term stability of the hydrophobic surface layer of molecules can be studied by monitoring the advancing contact angles over time. The values in the first 10 min are summarized for the selected samples in Figure 3.

As shown in Figure 3, it is true for all samples that the wettability is not constant (depending on the system, there is a decrease of 3–6°), but the significant differences between the systems remain even after 10 min. The improved wettability over time can be attributed to different processes: the reorganization of the surface polymer molecules [15] and/or penetration of water molecules into the layer. If the surfactant molecules were to desorb from the coating and dissolve in the droplet, the contact angle would approach the value measured on the CS coating (and could even decrease below it due to the lower surface tension of the dilute surfactant solution). Since there is no such decrease in contact angles during the 10-min period examined (although desorption can be instantaneous), the depletion of surfactants is probably hindered due to their binding to the positively charged CS molecules.

### 2.4. AFM Measurements: Surface Morphology

The morphology of the surface can also affect the measured contact angles. To study this effect, AFM images were taken of the selected most hydrophobic samples as well as the native CS coating for reference. The height images (which show the distance perpendicular to the plane of the substrate) and the phase images (which are related to the energetic heterogeneity of the surface) are summarized in Figure 4.

It can be seen from Figure 4 that the surface of the CS coating is smooth and energetically homogeneous; however, in the case of surfactant-containing coatings (especially the system impregnated with SDS), a different surface morphology can be observed, which shows the presence of small 30–50 nm domains (these presumably correspond to polymer–surfactant associates). Nevertheless, the surface roughness of the coatings (determined in 20 × 20 μm areas) is not significant. The Wenzel surface roughness factors (ratios between the actual and projected surface area, used in the Wenzel wetting model [34]) are below 1.001 in all cases, while the Sq surface roughness values (quadratic mean of profile height deviations from the mean line) are (in order from a to e) 2.3 nm, 2.0 nm, 4.1 nm, 5.6 nm, and 2.0 nm. This means that it is not the morphology but the chemical composition that significantly influences the measurable value of the contact angles. As an example, to validate the layer thickness values in Table 1, in the case of a “Gls/CS+SDBS (0.1 mM)” sample, the layer thickness was also determined by AFM (based on the depth profile taken from the scratched coating, see Figure A3 in Appendix A). The obtained value (357 ± 1 nm) is in good agreement with the layer thickness value in Table 1 (378 ± 10 nm).

### 2.5. XPS Measurements: Distribution of Surfactants in the Layer

To study the possible accumulation of the surfactants on the air–chitosan interfacial layer, the elemental composition of the nanolayers was examined by X-ray photoelectron spectroscopy. Since only the surfactants contain sulfur, their amount was characterized by the atomic percent of this element. The decomposition of the carbon and sulfur peaks and the peak analysis are presented in Appendix A (Figure A4). After measurement, a thin layer of the coating was removed with the argon cluster source; then, another spectrum was recorded. After that, another layer was removed, etc. Thus, the coatings were examined layer by layer from the surface to the substrate. We determined when the glass substrate was reached by the appearance of the silicon peak in the XPS spectra and the decrease in the sulfur peak close to zero. The sulfur content (in atomic percent) of the investigated most hydrophobic systems as a function of depth is shown in Figure 5. The depth values were calculated by dividing the layer thickness values in Table 1 by the number of etching cycles required to completely remove the coating (it was assumed that the same thickness of the coating layer was removed during each etching cycle).

It can be seen in Figure 5 that the sulfur content (sulfur atomic percent) of the pure CS coating is, not surprisingly, negligible in its entire depth (below 0.1%), and in the case of the mixed systems, the sulfur content in the bulk phase barely exceeds this low value. However, the amount of surfactant near the coating–air interface (ca. the outer 20 nm layer of the coatings) is an order of magnitude higher (0.7–0.9% for the mixed and 1.1–1.7% for the impregnated systems), which is a sign of surfactant adsorption at the interface due to their surface activity. A similar surface accumulation of phosphonated fatty acids was observed by Millet et al. with layer-by-layer SEM/EDX analysis [35]. For the impregnated systems, the sulfur atomic percent (and, therefore, the surfactant concentration) in the bulk phase is significantly higher (near 1%), and a similar interface enrichment can be observed, especially in the case of the SDBS-containing impregnated system, which shows the highest surfactant concentration on the surface of all samples (as well as the highest water contact angles, see Figure 2). While the amount of surfactants that can be added to the mixed systems is quite limited, a significant accumulation of surfactants occurred during impregnation, and the excess of surfactants in the interfacial layer is even greater.

### 2.6. Electrochemical Tests: Barrier Properties and Corrosion Protection

To characterize the barrier properties of the coatings in an aqueous medium, electrochemical measurements (polarization tests and electrochemical impedance spectroscopy) were performed. The results of these measurements are summarized in Figure 6, while the parameters obtained by the analysis of the data can be seen in Table 2.

The first measurement was the determination of the open circuit potential (OCP), which denotes the resting potential and the reference point for all other electrochemical tests. From Figure 6a, one can see that the equilibrium OCP value (after 30 min) of the CS samples and mixed systems is more positive than the value of the bare zinc, and the OCP value of impregnated systems even exceeds this. From this shift of OCPs to the more positive values, it can be concluded that fewer aggressive species reach the metal surface through the impregnated coatings. This arises from the property of covering layers, which effectively block the passage of electrolytes. The slight alternation of the OCP values during the 30-min measurement can be attributed to the inhomogeneity of the coatings or instabilities caused by local corrosion.

This finding is also confirmed by the other measurements. Figure 6b shows the Nyquist plot of the electrochemical impedance spectra of the samples, where the diameter of the upper semicircle correlates with the coating resistance and, consequently, with its barrier property. The largest loops belong to the impregnated coatings, which implies that these coatings can protect the metal surface against corrosive species. To gain more information about the corrosion process, an equivalent circuit model was fitted to the points of the impedance spectra (the fitted curves are plotted in Figure A5a in Appendix A). The figure of the equivalent circuit is presented in Figure 6b, while the fitted parameters are summarized in Table 2. Using the polarization resistance (*R_p_*) values, the inhibition efficiency (*IE*) of the coatings can be calculated with Equation (5).

Using the linear polarization measurements, the pseudo-porosity (*P*) of the coatings can be calculated with Equation (4). This value is above 90% for the CS coatings (Table 3) because there are many channels filled with unbound water in the coating, through which ions can easily reach the surface of the metal [15]. In the case of mixed systems, *P* is smaller, but a significant decrease can be observed for impregnated systems (the *p* value for Zn/CS/SDS (30 mM) is only 37%).

The cathodic branches of the polarization curves according to the Tafel representation curves exhibit a low slope, indicating a diffusion-controlled process. The slopes of the anodic branches (*b_a_*) are summarized in Table 3. The corrosion current densities (i_corr_) follow a similar trend to *P,* and the inhibition efficiency (*IE*) values calculated from them (see Equation (3)) are also the highest for the impregnated samples. In general, the *IE* values calculated through the two methods are close to each other.

The significantly better barrier properties of impregnated coatings are certainly to be found at their higher surfactant content. As already established based on the results in Figure 5, in the case of mixed systems, the surfactant concentration in the bulk phase is very low; hence, the only protective barrier is the adsorbed surfactant layer at the surface. On the other hand, in the case of impregnated systems, the amount of surfactant adsorbed at the surface is much higher (creating a stronger hydrophobic barrier), and the higher concentration of the surfactant in the bulk phase creates the possibility of the so-called micellar cross-linking. This means that several polymer chains can interact with the oppositely charged surface of the micelles formed in the hydrogel, and these micelles can thus function as network junctions, effectively cross-linking the polymer. Jiang et al. demonstrated this phenomenon in SDS-CS systems [37]. Since cross-linking can effectively enhance the barrier properties of CS coatings [38], the phenomenon can explain the obtained results.

### 2.7. Spectroscopic Ellipsometry: Swelling in a Humid Environment

To study the behavior of the coatings in a humid environment, ellipsometric measurements were carried out with water vapor in a chamber at 25 °C. The degrees of swelling on each pressure level were calculated using Equation (6) from the layer thickness values (obtained by fitting an optical model to the ellipsometric spectra). The calculated values are summarized in Figure 7, where the change in the relative pressure (sorption or desorption) is indicated by arrows on the lines.

The isotherm curve of the CS coating is sigmoid-shaped, similar to the type II adsorption isotherm curve with moderated sorption–desorption hysteresis. This curve is typical for chitosan-based systems [39,40]. In the case of the mixed systems, a similarly shaped curve can be observed, and the maximum degree of swelling is also not significantly different from the native system (ca. 75%), which once again highlights that the small amount of surfactant is not able to provide adequate barrier properties to the coatings. However, similar to the electrochemical measurements in an aqueous medium, impregnated systems showed the best performance out of all samples in a water vapor atmosphere, showing the lowest maximum swelling degrees and area of hysteresis. For the CS/SDBS system, the maximum swelling degree was reduced by a factor of 3 (ca. 25% instead of 75%). This can be attributed to the hydrophobic surfactant layer at the coating–air interface, as well as the micellar cross-linking, which has been shown to significantly reduce the degree of swelling of hydrogels [37].

## 3. Materials and Methods

### 3.1. Materials

Chitosan (medium molecular weight: 200,000–300,000 Da, degree of deacetylation: 75–85%, and viscosity of 1% solution in 1% aqueous acetic acid: 563 cP), sodium dodecyl sulfate (>99%, f.a.), and sodium dodecylbenzenesulfonate (technical grade) were provided by Merck (Darmstadt, Germany). Hydrochloric acid (37% f.a.) and sulfuric acid (96%, f.a.) were purchased from Carlo Erba (Cornaredo, Italy), isopropanol (99.7%, f.a.) was obtained from Reanal (Budapest, Hungary), acetic acid (99.8% f.a.) was purchased from Lachner (Neratovice, Czech Republic), and Na_2_SO_4_ (97%, f.a.) was purchased from Lachema (Brno, Czech Republic). All chemicals were used without any further purification. Water was purified by an Adrona system (Adrona Integrity+, Waterlab kft., Biatorbágy, Hungary) and had a specific electrical resistance of 18 MΩcm (ultrapure water). Microscope slides (Epredia, Breda, Netherlands) were used as glass substrates, and zinc plates (Bronzker Bt., Budapest, Hungary) were used as zinc substrates.

### 3.2. Coating Preparation

The coatings were deposited onto different substrates from the aqueous solution of CS by dip-coating. Glass was used for most of the measurements (UV-Vis spectroscopy, XPS, and wetting properties), and zinc substrates were used for electrochemical measurements. Glass (Gls) substrates were cleaned using 5 *w*/*w*% aqueous detergent solution, 10% H_2_SO_4_ solution, isopropanol, and ultrapure water. Zinc (Zn) substrates were prepared by wet grounding on SiC papers (P240-P4000, Buehler, Leinfelden-Echterdingen, Germany) and polished by a monocrystalline water-based diamond suspension (particle size of 3 μm, Buehler). Subsequently, the substrates were washed with isopropanol, 0.1 M HCl, and ultrapure water.

To create the coatings, solutions containing 1% and 1.5% CS were used. The 1% CS solution was prepared by dissolving 0.5 g CS in a mixture containing 500 μL of 99.8% acetic acid solution and 49 g of ultrapure water. The 1.5% CS solution was prepared by adding 0.75 g of CS to a mixture containing 500 μL of 98% acetic acid solution and 48.75 g of ultrapure water. The solutions were stirred for 24 h and then centrifuged for 30 min (4000 rpm, HERMLE Z36 HK, Dialab kft., Budapest, Hungary) to remove any insoluble CS particles.

Solutions of sodium-dodecylsulfate (SDS) and sodium dodecylbenzenesulfonate (SDBS) were made with different surfactant concentrations by dissolving the appropriate amount of surfactant powder in ultrapure water.

The surfactant-containing CS coatings were made using two methods. For the first method, a series of precursor solutions were prepared using a mixture of 1.5% CS and surfactant solution. In the precursor solutions, the CS concentration was kept constant at 1%, while the SDS and SDBS concentrations ranged from 0 mM to 1.5 mM and 0 mM to 0.75 mM, respectively (the upper limit of the surfactant concentration was the beginning of visible precipitation). The coatings were deposited by dip-coating with a device made by Plósz Mérnökiroda Kft. (Budapest, Hungary) at 25 °C, with a withdrawal speed of 5 cm/min. The coatings were dried for 24 h at room temperature. The systems created in this manner are referred to as ‘mixed systems’ and denoted as ‘Substrate/CS+Surfactant (concentration)’ (for example, the notation Gls/CS+SDBS (0.1 mM) refers to a coating on a glass substrate that is made from a solution containing 1% CS and 0.1 mM sodium-dodecybenzenesulfonate).

For the second method, surfactant-containing CS coatings were also made in two steps through impregnation: initially, coatings were made by surfactant-free 1% CS solutions using the dip-coating method. After drying, these coatings were impregnated in the concentrated solution of the surfactant (30 mM in the case of SDS and 1.5 mM in the case of SDBS) for the time period found to be optimal (30 min). These values were determined by preliminary experiments. After impregnation, the samples were carefully rinsed with ultrapure water and dried at room temperature. The systems created in this manner are referred to as ‘impregnated systems’ and denoted as ‘Substrate/CS/Surfactant (concentration)’.

### 3.3. UV-Vis Spectroscopy

The transmittance spectra of the samples were measured with an Analytik Jena Specord 200 UV-Vis spectrophotometer in the wavelength range of 350–1100 nm, with 1 nm resolution, and 10 nm/s measurement speed. The thickness and refractive index of the coatings were determined by a thin-layer optical model (which assumes transparent layers with an equal thickness on both sides of the substrate, see the Hild method [31]) fitted to the transmittance spectra. The optical model assumes one homogeneous layer with no absorption on each side of the substrate. The thickness and refractive index were calculated from the fitted values of six replicate samples (*n* = 6).

The effect of the surfactants on layer thickness was studied by calculating the relative increase in layer thickness (with respect to the pure CS coating). For this, Equation (2) was used, where *d* is the fitted thickness of the studied layer and *d_CS_* is the average thickness of the native CS layer.
(2)$$∆d_{rel}=\frac{d-d_{CS}}{d_{CS}}\cdot 100\;[\%]$$

### 3.4. Surface Tension Measurements

To obtain information about the polymer–surfactant interaction in the precursor solution, the surface tension of the pure and polymer-containing surfactant solutions was measured with the pendant drop method at 25 °C. The KRÜSS DSA30 device’s (Unitester kft., Veszprém, Hungary) ADVANCE 1.15.0 software calculated the surface tension by analyzing the shape of the droplet (the density of all solutions was considered to be the same as that of water). All surface tension values were calculated from the fitted data of 10 droplets (*n* = 10).

### 3.5. Wettability Measurements

To determine the wettability of the samples in order to obtain information about the possible accumulation of the surfactants at the layer-air interface, water contact angles were measured on the coatings deposited onto glass substrates. The measurement was carried out with a KRÜSS DSA30 device, using the sessile drop method at 25 °C with a humidity of at least 80% in the measurement chamber. Advancing contact angles were measured after the 10 μL droplet formation, while receding contact angles were measured after removing 5 μL of liquid from the droplet (the sessile drop method and drop build-up technique). Each value was determined from the contact angle data of at least 4 droplets (*n* = 8).

To study the time-dependent behavior of the wettability, changes in the advancing contact angles over time were measured. This involved building up 10 μL of the droplet and analyzing its contact angles over a period of 10 min. The data lines were calculated from the contact angle values of 2 droplets (*n* = 4).

### 3.6. Rheological Measurements

Rheological tests were performed with an Anton Paar Physica MCR301 (Anton Paar Hungary kft., Budapest, Hungary) device using a CP-25-1-SN54644 measuring head (Anton Paar Hungary kft., Budapest, Hungary) at 25 °C to obtain information about the dependence of viscosity on surfactant concentration. Shear stress was measured as a function of shear rate in the 0.1–100 s^−1^ range.

### 3.7. AFM Measurements

To study the surface morphology of the coatings, AFM images were recorded in tapping mode (AIST-NT SmartSPM 1000 AFM device and NanoSensors PPP-NCHR-20 needle with nominal radius <20 nm). Height and phase images were recorded in an area of 2 × 2 and 20 × 20 μm. In order to validate the layer thickness values obtained from optical measurements, in one case, the layer thickness of a sample was also determined with AFM in such a way that the sample was scratched; then, the depth profile was recorded perpendicular to the scratch. The data were evaluated with the Gwyddion 2.66 software [41].

### 3.8. Electrochemical Measurements

The barrier properties in a wet environment and the corrosion protection effect of the coatings applied on zinc substrates were investigated using electrochemical polarization tests. For the measurements, 0.2 g/L Na_2_SO_4_ solution was used as the electrolyte, which provided the corrosive medium. The measurements were carried out with a BioLogic SP-150 potentiostat (BioLogic, Grenoble, France) in a three-electrode cell, in which Ag/AgCl/KCl was used as the reference electrode, with an 80 × 50 mm Pt mesh as the counter electrode and the bare and coated Zn substrates as the working electrode.

The voltage range, where the entire polarization curve can be determined, was established by measuring the open circuit potential (OCP) for 30 min. To characterize the electrochemical system, Electrochemical Impedance Spectroscopy (EIS) was assessed in the frequency range of 10 mHz–10 kHz. Polarization measurements were carried out in the range of OCP ± 20 mV and OCP ± 250 mV.

The results of the polarization measurements in the wider region (OCP ± 250 mV) were plotted in a semi-logarithmic diagram (Tafel curves), and the corrosion current densities (*i_corr_*) were determined for each sample and for the bare Zn (*i*^0^*_corr_*). The inhibition efficiency (*IE*) was calculated from these values for each coating using Equation (3) [15].
(3)$$IE=\frac{i_{corr}^{0}-i_{corr}}{i_{corr}^{0}}\cdot 100\;[\%]$$

From the polarization measurements in the range of OCP ± 20 mV (linear polarization), the pseudo-porosity of the layers was determined by the following equation (Equation (4)) [42]:(4)$$P=\frac{R_{p_{s}}}{R_{p}}\cdot 10^{-\left|∆E_{corr}/b_{a}\right|}\cdot 100\;[\%]$$
where *R_ps_* and *R_p_* represent the polarization resistance of the bare and coated Zn (calculated from the slope of the linear polarization curves), respectively, Δ*E_corr_* is the difference in the corrosion potentials of the bare and coated substrate, and *b_a_* is the anodic Tafel slope of Zn (determined from the semi-logarithmic polarization curves).

To gain more information about the corrosion process, an equivalent circuit model was fitted to the Nyquist plot of the EIS spectra using the ZSimpWin 3.6 software. The accuracy of the fitting was evaluated using the chi-squared (χ^2^) value, which was approximately 10^−3^ or less. The polarization resistance (*R_p_*) values obtained from fitting provided an alternative method for calculating the inhibition efficiency (*IE*) using Equation (5) [43]:(5)$$IE=\frac{\left|Rp\right|-{\left|Rp\right|}_{0}}{\left|Rp\right|}\cdot 100\;[\%]$$
where ${\left|Rp\right|}_{0}$ is the polarization resistance value of the bare Zn sample and $\left|Rp\right|$ is the polarization resistance of the coated samples.

### 3.9. XPS Measurements

To obtain information about the depth distribution of sulfur-containing surfactants within the layers, XPS measurements were performed using a Thermo Scientific ESCALAB Xi^+^ instrument. The variation in the sulfur peak intensity in the XPS spectra is a widely used technique for studying polyelectrolyte coatings [44,45].

The samples were fixed to the sample holder using double-sided carbon tape, which ensured electrical contact. Considering that the sample is insulating, additional charge compensation was also necessary. The X-ray radiation was generated by the Kα radiation of an Al-anode, and the size of the X-ray spot was 900 μm. During the measurements, the samples were etched using a MAGCIS^TM^ argon ion source (Thermo Fischer SSC, Budapest, Hungary), worked in cluster mode (6000 eV, a cluster size of 1000, and an angle of incidence of 45° with respect to the surface normal) in a 3 mm area to reveal the component in-depth distribution. The evaluation of the data was performed using ThermoAvantage V5 software. The atomic concentrations (sulfur atomic percent, %) were obtained from the peak areas after removing the background by applying the sensitivity factor library (Althermo-1).

### 3.10. Spectroscopic Ellipsometry

To evaluate the vapor absorption characteristics of the coatings in a saturated water vapor atmosphere (and characterize their behavior in a humid environment), spectroscopic ellipsometry measurements were performed using a Semilab EP-12 instrument (Semilab Semiconductor Physics Laboratory Co. Ltd., Budapest, Hungary). After evacuating the sample chamber, the relative pressure of the water vapor was systematically increased and decreased between 0 and 1 in units of 0.1. At each pressure level, spectroscopic ellipsometric measurements were performed (wavelength range between 300 and 800 nm and angle of incidence of 60°) to investigate the swelling behavior of the coatings. The thickness and the refractive index of the layers were determined by fitting an optical model to the resulting spectra of Δ (phase difference between the p- and the s-polarized reflected light) and Ψ (amplitude ratio) using Semilab’s Spectroscopic Ellipsometry Analyzer (SEA v1.7.11) software. The swelling degree of the coatings was calculated with Equation (6), where *d_dry_* and *d_sw_* are the layer thickness values in dry and swollen states, respectively. During the calculations, constrained swelling of the coatings was assumed (i.e., it was assumed that the coating swells only in 1 dimension, which means that the layer thickness of the swollen coating is proportional to its volume), as previously confirmed for swelling in aqueous solutions [15].
(6)$$S=\frac{d_{sw}-d_{dry}}{d_{dry}}\cdot 100\;[\%]$$

## 4. Conclusions

Chitosan (CS) coatings containing sodium dodecyl sulfate (SDS) and sodium dodecylbenzenesulfonate (SDBS) were prepared in order to study the effect of the interfacial surfactant layer on the hydrophobicity and barrier properties. The surfactant was added to the system in two ways: by mixing it into the aqueous precursor solution (one-step method) and by the impregnation of the native CS layer from aqueous surfactant solutions. The one-step synthesis resulted in hydrophobic coatings, especially in the case of SDS additives. However, their barrier behavior was not suitable in an aqueous phase or in water vapor due to the low surfactant content in the layer.

In comparison, the barrier properties of the impregnated systems were better both in an aqueous medium and a water vapor atmosphere (smaller pseudo-porosity and hygroscopicity), which can be explained by the higher surfactant content in the layer. In this case, the SDBS additives showed better performance. The advancing values of the water contact angles reached 100°. The time-dependent water contact angles did not reveal significant deterioration of the self-assembled layer structure in any of the systems.

These coatings can be suitable, e.g., for corrosion protection of electronic devices in humid environments.

## Figures and Tables

**Figure 1 molecules-29-03111-f001:**
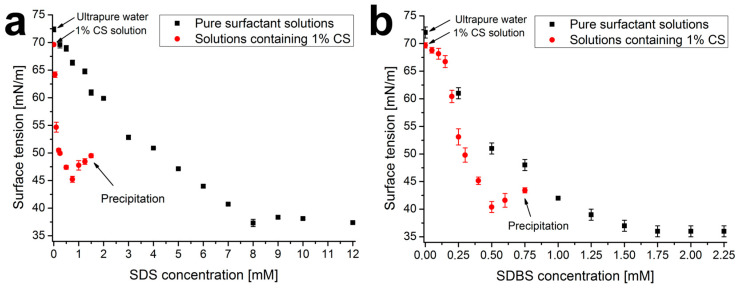
Surface tension of pure surfactant solutions (black squares) and SDS (**a**) and SDBS (**b**) solutions containing 1% CS (red circles), measured by the pendant drop method (mean ± standard deviation, *n* = 10).

**Figure 2 molecules-29-03111-f002:**
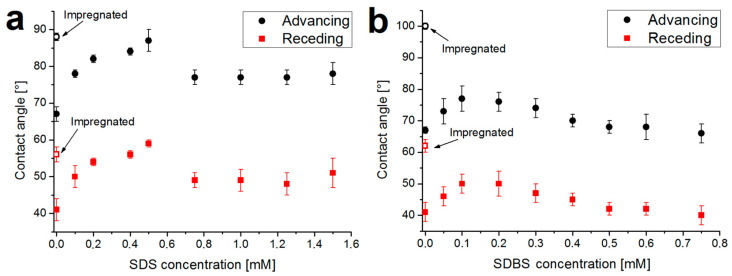
Advancing and receding water contact angles measured on coatings containing SDS (**a**) and SDBS (**b**). Impregnated coatings are marked with open markers at 0 concentration; the letters above the data points indicate the significant differences in the values (mean ± standard deviation, *n* = 8, *p* < 0.05).

**Figure 3 molecules-29-03111-f003:**
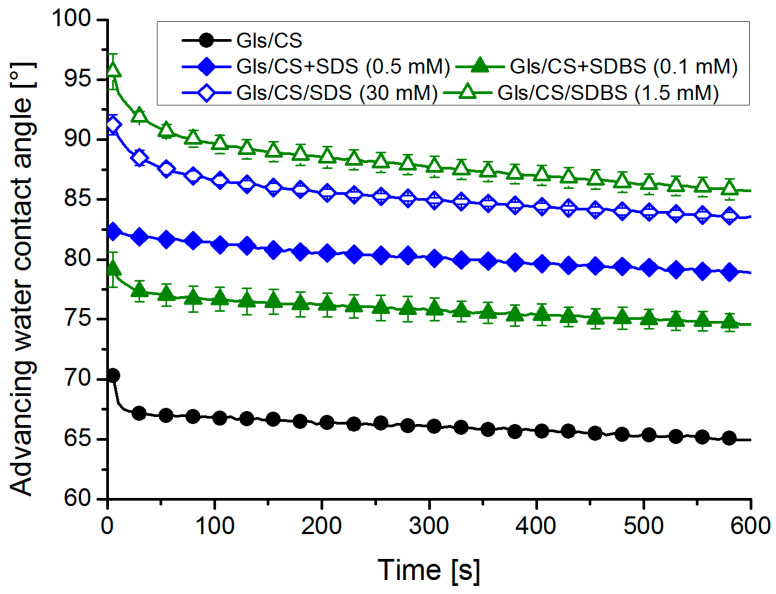
Time evolution of advancing water contact angles on the surface of the most hydrophobic coatings selected (solid markers: mixed systems and empty markers: impregnated systems, mean ± standard deviation, *n* = 6).

**Figure 4 molecules-29-03111-f004:**
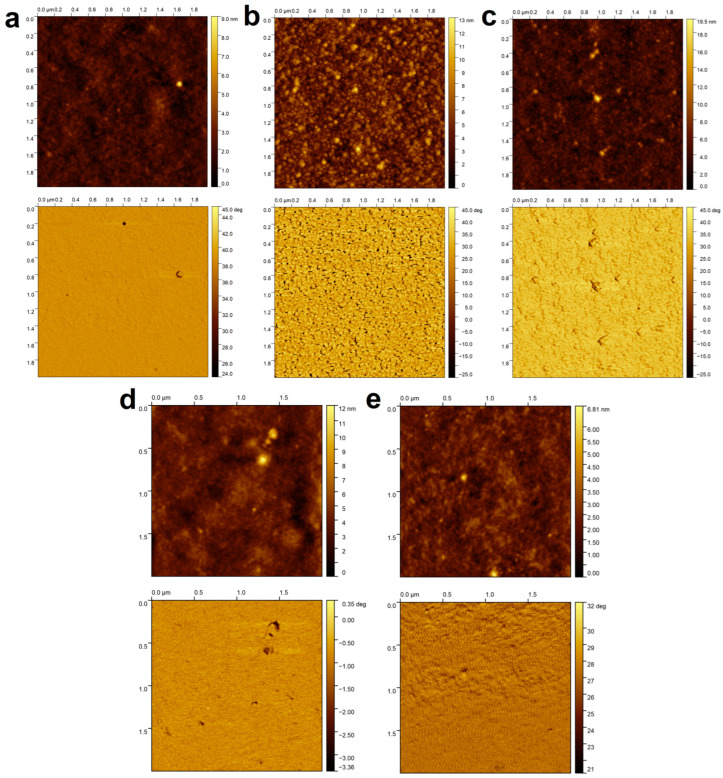
Height (up) and phase (down) images obtained by atomic force microscopy of the most hydrophobic samples on glass substrates selected ((**a**): Gls/CS, (**b**): Gls/CS/SDS (30 mM), (**c**): Gls/CS/SDBS (1.5 mM), (**d**): Gls/CS+SDS (0.5 mM), and (**e**): Gls/CS+SDBS (0.1 mM)).

**Figure 5 molecules-29-03111-f005:**
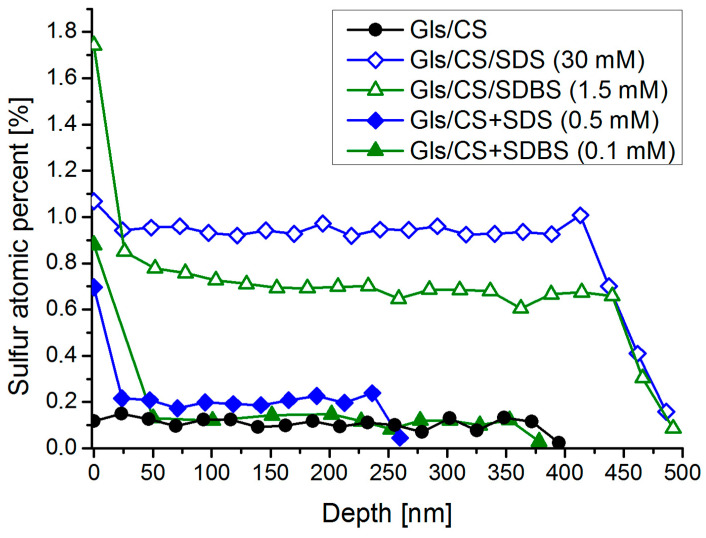
Sulfur atomic percent derived from XPS measurements along the depth of the samples for the most hydrophobic coatings selected (solid markers: mixed systems and empty markers: impregnated systems).

**Figure 6 molecules-29-03111-f006:**
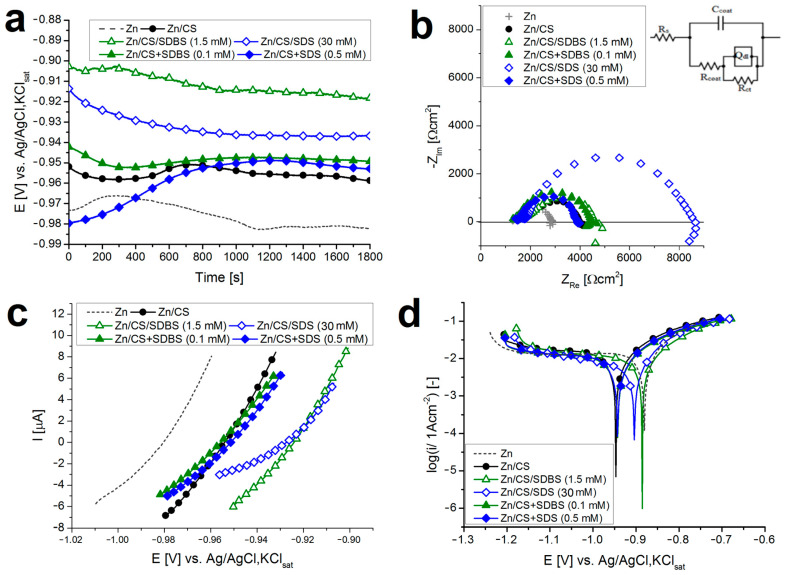
Results of the electrochemical measurements for the selected samples: determination of open circuit potential (**a**), Nyquist plot of electrochemical impedance spectra and the figure of the equivalent circuit fitted to the data, see Table 2. (**b**), linear polarization curves (**c**), and semi-logarithmic polarization curves (Tafel plots) (**d**). Solid markers: mixed systems and empty markers: impregnated systems.

**Figure 7 molecules-29-03111-f007:**
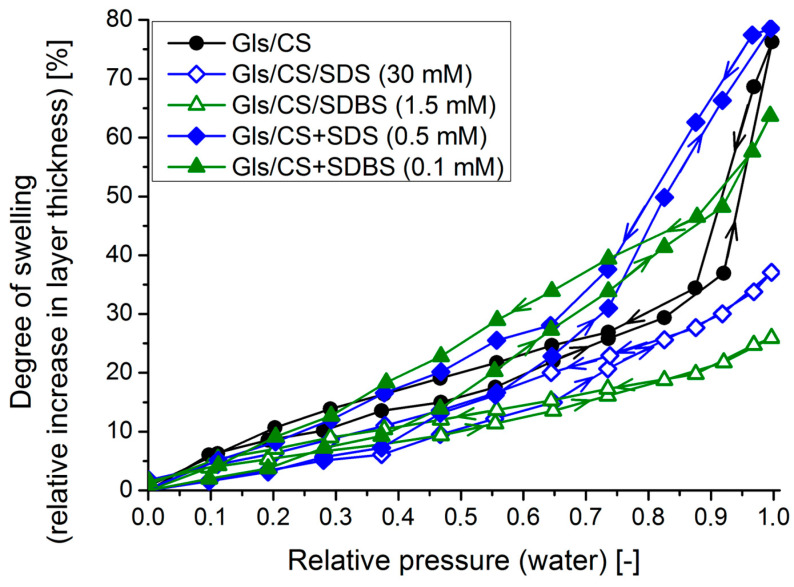
Calculated swelling degrees at different humidity values for the most hydrophobic coatings selected (solid markers: mixed systems and empty markers: impregnated systems, the arrows on the lines indicate the change of relative pressure during sorption and desorption).

**Table 1 molecules-29-03111-t001:** Refractive index and thickness values and Δ*d_rel_* values calculated from the thickness for the investigated systems (*n* = 6, Δ*d_rel_* values, obtained from thin-layer optical fitting, were calculated using Equation (2)).

Sample		Refractive Index (–)	Layer Thickness (nm)	Relative Increase in Layer Thickness (Δdrel) (%)
Native	Gls/CS	1.5329 ± 0.0028	395 ± 7	-
Mixed	Gls/CS+SDS (0.1 mM)	1.5417 ± 0.0008	325 ± 11	−18 ± 3
	Gls/CS+SDS (0.2 mM)	1.5419 ± 0.0010	310 ± 4	−22 ± 1
	Gls/CS+SDS (0.4 mM)	1.5478 ± 0.0010	293 ± 7	−26 ± 2
	Gls/CS+SDS (0.5 mM)	1.5486 ± 0.0011	260 ± 1	−34 ± 0
	Gls/CS+SDBS (0.05 mM)	1.5320 ± 0.0007	334 ± 11	−15 ± 4
	Gls/CS+SDBS (0.1 mM)	1.5346 ± 0.0019	378 ± 10	−4 ± 4
	Gls/CS+SDBS (0.2 mM)	1.5346 ± 0.0018	367 ± 11	−7 ± 4
	Gls/CS+SDBS (0.3 mM)	1.5304 ± 0.0015	409 ± 11	4 ± 5
Impregnated	Gls/CS/SDS (30 mM)	1.5238 ± 0.0027	486 ± 8	24 ± 2
	Gls/CS/SDBS (1.5 mM)	1.5321 ± 0.0026	492 ± 10	25 ± 2

**Table 2 molecules-29-03111-t002:** Results of fitting the Nyquist impedance plot (for the equivalent circuit, see Figure 6b). *R_s_* is the resistance of the electrolyte, and *C_coat_* and *R_coat_* are the capacitance and resistance of the coating, respectively, while Q_dl_ and R_ct_ are the pseudo-capacitance (constant phase element) and resistance of the electric double layer [36], respectively, and *χ*² characterizes the accuracy of the fitting. The *IE* values were calculated using Equation (5).

Sample	*R_s_* (kΩcm^2^)	*Q_coat_* (μSs^n^)	*R_coat_* (kΩcm^2^)	*Q_dl_* (μSs^n^)	*R_ct_* (kΩcm^2^)	*R_p_* = *R_coat_* + *R_ct_* (kΩcm^2^)	*χ²* (–)	*IE* (%)
Zn	1.29	-	-	24.79	1.58	1.58	6.51 × 10^−4^	-
Zn/CS	1.43	0.39	0.19	16.76	2.42	2.61	2.50 × 10^−3^	39.5
Zn/CS+SDS (0.5 mM)	1.22	0.08	0.37	14.33	3.11	3.35	6.35 × 10^−4^	52.8
Zn/CS+SDBS (0.1 mM)	1.21	0.10	0.23	16.54	2.97	3.20	7.77 × 10^−4^	52.4
Zn/CS/SDS (30 mM)	1.49	0.12	0.33	17.31	6.74	7.07	1.01 × 10^−3^	77.7
Zn/CS/SDBS (1.5 mM)	1.39	0.12	0.45	22.94	2.86	3.32	2.49 × 10^−3^	50.6

**Table 3 molecules-29-03111-t003:** Kinetic parameters (*b_a_*, *i_corr_*, and *E_corr_*), polarization resistance *R_p_*, and pseudo-porosity *P* determined from the potentiodynamic polarization of the most hydrophobic samples selected.

Sample	Linear Polarization			Tafel Interpretation			
	*E_corr_* (V)	*R_p_* (Ω)	*P* (%)	*ba* (V/dec)	*E_corr_* (V)	*i_corr_* (mA/cm^2^)	*IE* (%)
Zn	−0.985	3206 (R_ps_)	-	0.167	−0.880	13.93	-
Zn/CS	−0.970	3240	94	0.258	−0.959	8.34	40.2
Zn/CS+SDS (0.5 mM)	−0.954	4225	68	0.230	−0.942	9.20	33.9
Zn/CS+SDBS (0.1 mM)	−0.955	3954	73	0.304	−0.942	6.16	55.8
Zn/CS/SDS (30 mM)	−0.925	6897	37	0.334	−0.903	5.38	61.4
Zn/CS/SDBS (1.5 mM)	−0.911	4690	52	0.214	−0.885	6.31	54.7

## Data Availability

The data are available upon request.

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
