# Peer review of "Chitosan–Surfactant Composite Nanocoatings on Glass and Zinc Surfaces Prepared from Aqueous Solutions"

_molecules, 2024, doi:10.3390/molecules29133111_

Round 1

Reviewer 1 Report

Comments and Suggestions for Authors

Péter Márton proposes a study on the preparation of chitosan films with surfactants to produce coatings with possible application as corrosion protectives. The topic of the paper is of scientific interest, although the author should address the following important aspects.

1)      It is not clear why the author is comparing systems at different concentrations of surfactants for each measurement. For example, it is not clear why the sample Gls/CS/SDS (30 mM) has one order of magnitude larger concentration compared to Gls/CS/SDBS (1.5 mM). Every characterization presents different samples at different concentrations and therefore it is not possible to make direct conclusions on efficiencies as coatings. The author should be able to extensively explain the choice in the concentrations, or otherwise compare analysis at equal concentrations of surfactants.

2) What is the relevance of the study in the field of application? The author mentions the application in the “protection for metals in maritime transport”. In this case, the coating would be immersed in water. Therefore, one would expect that the surfactant is depleted from the chitosan coating. Moreover, does the author expect the coating to maintain its adhesion to the substrate in case of high humidity/immersion in water? These aspects should be discussed.

3)      Does the author expect a partial depletion of the surfactant during its use as coating in corrosive environment?

4)      The author should improve the discussion of the data. No numbers are reported in the discussion, but only trends. Please improve this aspect for each figure (i.e. discussion on Figure 2: no numbers of advancing and receding contact angles are explicitly mentioned in the text).

5)      Figure 2. How does the author discriminate from significant differences in data values assigning letters? The reader should be able to discriminate based on the standard deviation. Consider if not relevant to eliminate lettering.

6)      If Eq2. was used to calculate the thickness, what is the use of Eq.1? Please explain.

7)      Please add more relevant information relatively to the characterization of chitosan. At least the viscosity should be provided.

8)      Line 182-185. The reason why the author hypothesizes that impregnated are similar to 0.55mM it’s not clear.

9)      Line 224. Please explain the term “odd surface morphology”.

10)   Line 302-306. The concept of micellar cross-linking is not clearly stated.

11)   The English expression needs improvements.

Please consider also minor comments.

12)   CS / chitosan. Please be consistent with acronyms.

13)   Line 246. Wrong comma.

14)   Line 274. Comma in the wrong position.

Comments on the Quality of English Language

 The English expression needs improvements.

Author Response

The authors are grateful for the opportunity to improve the manuscript based on the Editors and Reviewers' comments. The answers to the questions and comments and the list of changes can be found below. The comments are in bold, and the responses and list of changes is provided for each comment. The changes were marked in the manuscript and appendix in red, and the deleted parts are red and crossed out. A clean manuscript is also provided, without the deleted parts and the red marks. Whenever a line is referred to in this document, it refers to the marked version of the manuscript.

Péter Márton proposes a study on the preparation of chitosan films with surfactants to produce coatings with possible application as corrosion protectives. The topic of the paper is of scientific interest, although the author should address the following important aspects.

Comment #1:

It is not clear why the author is comparing systems at different concentrations of surfactants for each measurement. For example, it is not clear why the sample Gls/CS/SDS (30 mM) has one order of magnitude larger concentration compared to Gls/CS/SDBS (1.5 mM). Every characterization presents different samples at different concentrations and therefore it is not possible to make direct conclusions on efficiencies as coatings. The author should be able to extensively explain the choice in the concentrations, or otherwise compare analysis at equal concentrations of surfactants.

Answer #1:

The authors thank the Reviewer the observation. In the case of the mixed samples (/CS+SDS and /CS+SDBS) precursor solutions with different concentrations were created, and the best systems were selected based on water contact angles (assuming that the more hydrophobic surface corresponds with better barrier properties). For the impregnated systems (/CS/SDS and /CS/SDBS), the effect of time and surfactant concentration were investigated on the water contact angles during preliminary experiments, and the best systems were selected according to this (which were the  30 mM solution for SDS and the 1.5 mM solution for SDBS and 30 minute impregnation time). These results were not presented in the article for the sake of compactness. In addition, in the case of impregnation, the concentration of the solution and the concentration of the surfactants in the coating do not match, the goal was simply to accumulate as many surfactants as possible in the coating and its boundary layer.

However, to explain the concentrations used, the manuscript has been expanded.

Changes:

„Nanolayers were formed from pure and surfactant-containing CS solutions using the dip-coating technique. Additionally pure CS layers were impregnated in more concentrated surfactant solutions (30 mM SDS and 1.5 mM SDBS) for 30 minutes as a different modification technique (these values were determined based on previous preliminary experiments, which are not detailed here). To study the effect of the surfactant content on the properties of the layers, the thickness and refractive index were measured…” (Line 137)

„After drying, these coatings were impregnated in the concentrated solution of the surfactant (30 mM in the case of SDS, 1.5 mM in the case of SDBS) for the time found to be optimal (30 minutes). These values were determined by preliminary experiments.” (Line 427)

Comment #2:

What is the relevance of the study in the field of application? The author mentions the application in the “protection for metals in maritime transport”. In this case, the coating would be immersed in water. Therefore, one would expect that the surfactant is depleted from the chitosan coating. Moreover, does the author expect the coating to maintain its adhesion to the substrate in case of high humidity/immersion in water? These aspects should be discussed.

Answer #2:

When the introduction refers to "temporary corrosion protection for metals in maritime transport", it refers to the corrosion protection of metal parts transported on ships, which must primarily be protected against the corrosive effects of humid, salty air, and are not directly immersed in water. The situation is similar in the case of electronic circuits. The introduction has been modified to emphasize this. Long-term soaking in an aqueous medium during electrochemical tests is more of a measurement technique than an application condition (electrochemical measurements can only be performed in a liquid medium). About the possible depletion of surfactants please see the next answer.

Changes:

„The coatings prepared by this way can be used in wide range of applications, for example temporary corrosion protection for metals in maritime transport (against humid, salty air) or to protect electronic circuits in humid circumstances.” (Line 101)

“To evaluate the vapour absorption characteristics of the coatings in saturated water vapour atmosphere (and characterize their behaviour in humid environment), spectroscopic ellipsometry measurements were performed using a Semilab EP-12 instrument.” (Line 533)

Comment #3:

Does the author expect a partial depletion of the surfactant during its use as coating in corrosive environment?

Answer #3:

The authors thank the Reviewer this question. When anionic surfactants interact with cationic polyelectrolyte (chitosan), the electrostatic forces can stabilize the structure of the coating, preventing the depletion of surfactants in aqueous environment. This is supported by the observation that, in the case of mixed systems, when a high surfactant-concentration was used, a water-insoluble precipitate was formed (this limited the amount of surfactant that could be mixed in). In addition, the examination of the time-dependence of the contact angles (Fig. 3) also shows that the structure is stable and surfactants do not dissolve from the coating (if this were to happen, the polarity of the surface of the coating would increase significantly, and the surface tension of the liquid due to the dissolved surfactants would also decrease, which would drastically reduce the contact angle, below the value measured on the CS coating). Since the dissolution and rearrangement is a very fast process, and such a decrease of the contact angles cannot be experienced in the first 10 minutes, the depletion of surfactants is probably negligible. These points are now included in the manuscript.

Changes:

“If the surfactant molecules were to desorb from the coating and dissolve in the droplet, the contact angle would approach the value measured on the CS coating (and could even decrease below it due to the lower surface tension of the dilute surfactant solution). Since there is no such decrease in contact angles during the 10-minute period examined (although desorption can be instantaneous) the depletion of surfactants is probably hindered due to their binding to the positively charged chitosan molecules.” (Line 220)

Comment #4:

The author should improve the discussion of the data. No numbers are reported in the discussion, but only trends. Please improve this aspect for each figure (i.e. discussion on Figure 2: no numbers of advancing and receding contact angles are explicitly mentioned in the text).

Answer #4:

The authors are thankful for this suggestion. The “Results” section has been supplemented with numeric data in the text for better readability. 

Changes:

„Fig. 1 shows that chitosan by itself reduces slightly the surface tension of solutions compared to pure water (from 72 mN/m to 69 mN/m), due to its weak surface activity.” (Line 118)

„The upper limit of the measurement was the aggregation of these complexes in both cases, where visible precipitation was observed (1.5 mM for SDS and 0.75 mM for SDBS).” (Line 130)

“The ca. 25% increase in layer thickness in the case of impregnated systems can be explained by the increase in volume due to surfactant molecules accumulating in the coating during impregnation.” (Line 169)

“In the case of mixed systems (solid markers in Fig. 2), the advancing and receding contact angles increase at low surfactant concentrations (by ca. 20° for SDS and 10° for SDBS).” (Line 186)

„However, when the surfactant concentration increases further, the surface becomes more polar again (the advancing contact angles decrease to 80° and 70°).” (Line 189)

„For the coatings impregnated with SDS (open markers in Fig. 2a), the contact angles are roughly the same (85-90°) as in the case of the most hydrophobic mixed system (0.5 mM).” (Line 195)

“As shown in Figure 3, the wettability is not constant in either case (depending on the system, there is a decrease of 3-6°), but the significant differences between the systems remain even after 10 minutes.” (Line 217)

„It can be seen in Fig. 5 that the sulfur content (sulfur atomic percent) of the pure CS coating, not surprisingly, is negligible in its entire depth (below 0.1%),…” (Line 277)

„However, the amount of surfactant near to the coating-air interface (ca. the outer 20 nm layer of the coatings) is an order of magnitude higher (0.7-0.9% for the mixed, and 1.1-1.7% for the impregnated systems), which is a sign of surfactant adsorption at the interface due to their surface activity.” (Line 280)

„For the impregnated systems, the sulfur atomic percent (therefore the surfactant concentration) in the bulk phase is significantly higher (near 1%)” (Line 285)

„In the case of the mixed systems, the shape of the curve is similar and the maximum degree of swelling is not significantly different from the native system (ca. 75%),” (Line 371)

Comment #5:

Figure 2. How does the author discriminate from significant differences in data values assigning letters? The reader should be able to discriminate based on the standard deviation. Consider if not relevant to eliminate lettering.

Answer #5:

The authors thank the Reviewer for pointing this out. It is true, that the marks of the standard deviation should be the first guideline to discriminate the differences between the data points. Although nowadays (especially in the case of biochemical research) the so-called “compact letter display” is often used to indicate significant differences. This means that the significant difference between the data points is investigated with ANOVA tests and then marked with letters. In doing so, the data are marked according to decreasing size, during which the data points that are not significantly different are given the same, while the data that are significantly different are given different letters. If two data points do not differ significantly (p<0.05), they have the same letter in their designation.

However, since the statistical test in the manuscript could not be performed for all data, so letters are only used in Figure 2. For the sake of clarity, they were removed together with the subsection discussing statistical tests.

Changes:

Letters were removed from Fig. 2.

Section 3.11 was removed.

Comment #6:

If Eq2. was used to calculate the thickness, what is the use of Eq.1? Please explain.

Answer #6:

The authors thank the question. Equation 1. is the Landau-Levich equation, which estimates the thickness of the liquid film formed on the substrate during dip-coating as a function of the parameters of the coating deposition. Since the final coating is formed by drying of the liquid film, the final thickness is not given by the relation, but it helps to illustrate the effect of different parameters (e.g. viscosity: increasing viscosity of precursor solutions causes increasing liquid film thickness, thus probably increasing layer thickness).

The real thickness of the coatings were determined by fitting a thin layer optical model to the transmittance spectra of the coatings, described in Ref 31. In order to visualize the effect of surfactant additives on the layer thickness, the relative increase of layer thickness was calculated from the obtained thickness with respect to the value of the unmodified chitosan coating (this process was described by equation 2).

To make this more clear the sections dealing with these equations were expanded, and an additional column were added to Table 1. with the fitted layer thickness values (from which the relative increase values were calculated with Eq. 2.) Additionally, two examples for the fitting are now presented in the Appendix, on Fig. A1.

Changes:

„To study the effect of the surfactant content on the properties of the layers, the thickness and refractive index were measured by UV-Visible spectroscopy, fitting a thin-layer optical model to the transmittance spectra (Hild-model, see Ref [31]).” (Line 142)

„Refractive index and thickness values” (Line 151)

„The Landau-Levich equation can be used to interpret the trends in the layer thickness values, which estimates the thickness of the liquid film.” (Line 156)

“The ca. 25% increase in layer thickness in the case of impregnated systems can be explained by the increase in volume due to surfactant molecules accumulating in the coating during impregnation.” (Line 169)

“The thickness and refractive index of the coatings were determined by a thin-layer optical model, (which assumes transparent layers with equal thickness on both sides of the substrate see Hild-method [31]) fitted to the transmittance spectra.” (Line 437)

One extra column was added to Table 1. with the fitted layer thickness data.

Examples for the model fitting to the transmittance spectra was added to Fig. A1.

Comment #7:

Please add more relevant information relatively to the characterization of chitosan. At least the viscosity should be provided.

Answer #7:

The authors are thankful for this suggestion. The viscosity data provided by the manufacturer was added to the “materials” section.

Changes:

„Chitosan (Medium molecular weight: 200.000-300.000 Da, Degree of deacetylation: 75-85%, viscosity of 1% solution in 1% aqueous acetic acid: 563 cP),…” (Line 383)

Comment #8:

Line 182-185. The reason why the author hypothesizes that impregnated are similar to 0.55mM it’s not clear.

Answer #8:

The authors thank the Reviewer this observation. The contact angles are primarily influenced by the chemical composition (and possibly the morphology) of the surface layer of the coatings, and even a monomolecular layer of a compound can significantly change the hydrophobicity of the surface. This means that if the same contact angles can be measured in the case of mixed and impregnated systems, it is likely that the hydrophobicity-causing surface structure of the SDS molecules is also the same. Since this is the value for the most hydrophobic samples, we assume that this structure is the full coverage of SDS molecules. The manuscript were modified to clarify these findings.

Changes:

„For the coatings impregnated with SDS (open markers in Fig. 2a), the contact angles are roughly the same (85-90°) as in the case of the most hydrophobic mixed system (0.5 mM). The reason for this may be that, in both cases, the surface of the coatings is fully covered with SDS molecules, and this hydrophobic surface layer determines the contact angle value.” (Line 197)

Comment #9:

Line 224. Please explain the term “odd surface morphology”.

Answer #9:

When the text refers to “odd surface morphology”, we intended to indicate that the morphology is different compared to the unmodified coating, which means small domains corresponding to polymer surfactant associates. The manuscript has been amended to clarify this meaning.

Changes:

„…however, in the case of surfactant-containing coatings (especially the system impregnated with SDS), a different surface morphology can be observed, which shows the presence of small 30-50 nm domains (these presumably correspond to polymer-surfactant associates).” (Line 242)

Comment #10:

Line 302-306. The concept of micellar cross-linking is not clearly stated.

Answer #10:

The authors are thankful for this observation. Micellar cross-linking can occur in polyelectrolyte hydrogels containing oppositely charged surfactants at a sufficiently high concentration. In this case, the surfactants (electrostatically connected to the polymer chains) assemble into micelle-like structures, and since several polymer chains can be connected to the surface of a formed micelle in this way, the micelles can function as network junctions, cross-linking the polymer. The structure formed in the bulk phase of the coating can also be seen in the figure on the right side of the graphical abstract.

Changes:

„…so-called micellar cross-linking. This means that several polymer chains can interact with the oppositely charged surface of the micelles formed in the hydrogel, and these micelles can thus function as network junctions effectively cross-linking the polymer.” (Line 351)

Comment #11:

The English expression needs improvements.

Answer #11:

The authors thank the Reviewer this advice. The manuscript was read through and corrected several places.

Changes:

Wording and spelling errors were corrected in lines 35, 39, 40, 46, 48, 54, 55, 56, 61 and numerous other places.

Please consider also minor comments.

Comment #12:

CS / chitosan. Please be consistent with acronyms.

Answer #12:

The authors thank this suggestion. The acronym was corrected in the appropriate places.

Changes:

“Chitosan” was changed to “CS” in lines 56, 59, 60, 61 and several other places.

Comment #13:

Line 246. Wrong comma.

Answer #13:

The authors are thankful for this observation. The comma was deleted.

Changes:

„It can be seen in Fig. 5 that the sulfur content (sulfur atomic percent)…” (Line 276)

Comment #14:

Line 274. Comma in the wrong position.

Answer #13:

The authors thank the Reviewer this observation. The comma was deleted.

Changes:

„From Fig. 6a one can see that the equilibrium OCP value…” (Line 304)

Comments on the Quality of English Language

The English expression needs improvements.

Reviewer 2 Report

Comments and Suggestions for Authors

The manuscript entitled “Chitosan-surfactant composite nanocoatings prepared from aqueous solutions” reports on the use of a composite of chitosan and surfactant SDS and SDBS as a coating for glass and zinc substrates. The coatings were fabricated by two different methods and both methods showed to affect the barrier properties of the coating.  Albeit several points explored by the authors are of great interest, some concerns have to be addressed before consideration for publication:

1.       The manuscript focuses on coatings for glass and zinc substrates, and this should be included in the title.

2.       The UV-Vis spectrum in Fig. A1 of the case 0.5mM SDS appears to be different from other concentrations and the peak position is shifted. Any explanation? Also details on how thickness was determined from the UV data should be given. The fitted data should be plotted.

3.       XPS: the fitted XPS spectra should be presented even in a supporting information. It isn’t clear for me how the bulk concentration of sulfur was determined up to 350 nm using XPS although XPS instrument can reach only analysis depth of 10-20 nm!

4.       The introduction should be strengthened by including relevant previous works on eco-friendly corrosion inhibition coatings such as the book: corrosion protection of metals and alloys using graphene and biopolymers.. and the references 10.1149/1945-7111/acfa69, 10.1016/j.ijbiomac.2023.124787, 10.1016/j.jallcom.2023.173307

5.       In all figures, the decimal in numbers should be corrected to dot according to the standard format.

6.       In Fig 6a, the OCP isn’t constant and alters high and low with time. Any explanation for such wavy trend in the OCP values?

7.       Fig 6b, the 10 mM SDS semicircle is larger than the 1.5 mM SDBS, however a contradictory behavior was observed in the OCP values. Could you repeat this measurement or rationalize this different behavior.

8.       Authors are advised to fit the Nyquist plots using a suitable equivalent circuit and compare the resulting charge transfer resistance of the cases and use it to calculate the IE and add them in Table 2. You can refer to the following paper in your discussion (Phosphate glass doped with niobium and bismuth oxides as an eco-friendly corrosion protection matrix of iron steel in HCl medium…)

9.       Also, it is advised to determine anodic and cathodic Tafel slopes from Fig. 6d and discuss the values to give insights into the corrosion determining reaction (anodic or cathodic). You can take guide from (Electropolymerized conducting polyaniline coating on nickel-aluminum bronze alloy for improved corrosion resistance in marine environment)

10.   Typo errors e.g. L54 (and cosmetics), avoid conjunctions as (so..),

Comments on the Quality of English Language

minor editing

Author Response

The authors are grateful for the opportunity to improve the manuscript based on the Editors and Reviewers' comments. The answers to the questions and comments and the list of changes can be found below. The comments are in bold, and the responses and list of changes is provided for each comment. The changes were marked in the manuscript and appendix in red, and the deleted parts are red and crossed out. A clean manuscript is also provided, without the deleted parts and the red marks. Whenever a line is referred to in this document, it refers to the marked version of the manuscript.

Comments and Suggestions for Authors

The manuscript entitled “Chitosan-surfactant composite nanocoatings prepared from aqueous solutions” reports on the use of a composite of chitosan and surfactant SDS and SDBS as a coating for glass and zinc substrates. The coatings were fabricated by two different methods and both methods showed to affect the barrier properties of the coating.  Albeit several points explored by the authors are of great interest, some concerns have to be addressed before consideration for publication:

Comment #1:

The manuscript focuses on coatings for glass and zinc substrates, and this should be included in the title.

Answer #1:

The authors thank the Reviewer for pointing this out. The manuscript title was modified according to the suggestion.

Changes:

The new title is “Chitosan-surfactant composite nanocoatings on glass and zinc surfaces prepared from aqueous solutions”.

Comment #2:

The UV-Vis spectrum in Fig. A1 of the case 0.5mM SDS appears to be different from other concentrations and the peak position is shifted. Any explanation? Also details on how thickness was determined from the UV data should be given. The fitted data should be plotted.

Answer #2:

The authors are thankful for this comment. The thickness and refractive index values of the coatings were determined by fitting a thin-layer optical model to the transmittance spectra. Due to thin-layer interference, maxima and minima appear on the transmittance spectra, from the position and magnitude of which (knowing the appropriate physical relationships) the thickness and refractive index of the coatings can be determined. The model used is called the Hild-model, which assumes transparent layers of equal thickness on both sides of the transparent substrate. A detailed description of the model can be found in Ref. 34 (Hild et al., 2007). The text in the experimental section was expanded with the short description of the model. Examples for the fitting are also added to Fig. A1.

As the concentration of the surfactants increase in the precursor solution, the polymer and surfactant molecules form increasingly larger associates. Above a certain concentration of surfactants (1.5 mM for SDS and 0.75 mM for SDBS), their precipitation can also be observed, but even at lower concentrations they cause opalization of the precursor solutions. This effect is further strengthened during layer formation due to the concentration of the solution, causing strong light scattering of the coatings. Since the light scattering is more intense for shorter wavelengths, in these cases a strong decrease in transmittance can be seen in the range below 500 nm on the spectra. The light scattering above a surfactant concentration will be so large that it will mask the maxima and minima due to interference, making it impossible to fit the thin layer optical model. The first sign of this scattering is causing the decrease in the transmittance of the “Gls/CS+SDS(0.5 mM)” sample. The manuscript now includes the short summary of these findings.

Changes:

„To study the effect of the surfactant content on the properties of the layers, the thickness and refractive index were measured by UV-Visible spectroscopy, fitting a thin-layer optical model to the transmittance spectra (Hild-model, see Ref [31]).” (Line 143)

“The thickness and refractive index of the coatings were determined by a thin-layer optical model, (which assumes transparent layers with equal thickness on both sides of the substrate see Hild-method [31]) fitted to the transmittance spectra.” (Line 437)

“In the case of higher surfactant concentrations, the significant light scattering of the coatings did not allow the evaluation with the Hild-model (due to the light scattering, the transmittance at shorter wavelengths decreases significantly, the first sign of this can be seen on the “Gls/CS+SDS(0.5 mM)” sample spectrum in Fig. A1a).” (Line 146)

“Fig. A1a and b show examples for the transmittance spectra of the coatings made from precursor solutions containing SDS and SDBS in different concentrations. The slight light scattering of the more concentrated solutions resulted in the decrease of transmittance at shorter wavelengths (see the spectrum of “Gls/CS+SDS(0.5 mM)” sample). The layer thickness and refractive index values were calculated by fitting the Hild-model [33] to the spectra. Two examples for the fitting are in Fig. A1c (the R2 value were above 0.99 in all cases).” (Line 592)

Comment #3:

XPS: the fitted XPS spectra should be presented even in a supporting information. It isn’t clear for me how the bulk concentration of sulfur was determined up to 350 nm using XPS although XPS instrument can reach only analysis depth of 10-20 nm!

Answer #3:

The authors are thankful for this suggestion. Examples of the peaks of carbon and sulfur from the XPS spectra are now included in the Appendix with the description of the chemical nature (chemical bonds) of these two elements in the material (obtained by peak decomposition). XPS provides information of the upper 5-10 nm, if one intends to investigate deeper regions, argon sputtering can be used. This means that the upper layer of the material can be etched away with a beam of argon ions (or for softer materials, with argon clusters), thus revealing the deeper regions, from which a new spectrum can be taken (from the upper 5-10 nm layer). After another etching a new deeper region is brought to the surface until the entire thickness of the coating can be analyzed. A short description of this process in now presented in the manuscript.

Changes:

„The decomposition of the carbon and sulfur peaks and the peak analysis is presented in the Appendix (Fig. A4). After a measurement, a thin layer of the coating was removed with the argon cluster source, then another spectra was recorded, after that another layer was removed etc. That way the coatings were examined layer-by-layer from the surface to the substrate.” (Line 259)

„The depth values ​​were calculated by dividing the layer thickness values ​​in Table 1. by the number of etching cycles required to completely remove the coating (it was assumed that the same thickness of coating layer was removed during each etching cycle).” (Line 266)

„Examples for the decomposition of the carbon and sulfur peaks in the case of a “Gls/CS+SDS (0.5 mM)” sample are presented in Fig. A4. The peak fitting was performed in CasaXPS software. The binding energy was set by fixing the C–(C,H) component of the C 1s peak at 284.8 eV. The carbon high resolution spectra were decomposed by Lorentzian Asymmetric Lineshape (LA) to the following 4 components: at 284.8 eV, typical of carbon only bound to carbon and hydrogen; near 286.3 eV, typical of carbon making a single bond with oxygen or nitrogen; near 287.8 eV typical of acetal and amide; near 288.8 eV typical of carboxyl group (all of the above are characteristic of chitosan) [46,47]. The sulfur spectra were decomposed by GL(30) peak shape (Gaussian-Lorentzian product function with a mixing parameter of 30). The splitting value for 2p3/2 and 2p1/2 was 1.2 eV. The area ratio between the 2p 3/2 and 2p 1/2 peaks was set to be 2:1. The binding energy belonging to 169.8 eV and 168.6 eV can be attributed to SO4 species presented in the surfactant [48,49].” (Line 627)

Comment #4:

The introduction should be strengthened by including relevant previous works on eco-friendly corrosion inhibition coatings such as the book: corrosion protection of metals and alloys using graphene and biopolymers.. and the references 10.1149/1945-7111/acfa69, 10.1016/j.ijbiomac.2023.124787, 10.1016/j.jallcom.2023.173307

Answer #4:

The authors thank the Reviewer this comment. The suggested book and two of the articles are included in the introduction as Ref 4, 11 and 12.

Comment #5:

In all figures, the decimal in numbers should be corrected to dot according to the standard format.

Answer #5:

The authors are thankful for this observation. The figures were corrected.

Changes:

The comma were corrected to dot on Figures 1, 2, 5, 6, 7 and A1.

Comment #6:

In Fig 6a, the OCP isn’t constant and alters high and low with time. Any explanation for such wavy trend in the OCP values?

Answer #6:

The authors thank the Reviewer this question. During the 30-minute OCP measurement the slowly establishing equilibrium can be disturbed by the surface's inhomogeneity and the electrolyte's aggressiveness. It also can reflect a certain instability of the system, caused maybe by local corrosion. These findings are briefly included in the manuscript.

Changes:

„The slight alternation of the OCP values during the 30-minute measurement can be attributed to the inhomogeneity of the coatings or instabilities caused by local corrosion.” (Line 309)

Comment #7:

Fig 6b, the 10 mM SDS semicircle is larger than the 1.5 mM SDBS, however a contradictory behavior was observed in the OCP values. Could you repeat this measurement or rationalize this different behavior.

Answer #7:

The authors are thankful for this suggestion. During the original measurements, two samples of each type were measured, but the data of only one is shown in Figure 6. The OCP value of the second “Zn/CS/SDS (30 mM)” sample measured after 30 minutes is more positive, but still below the value of “Zn/CS/SDBS (1.5 mM)”. A possible explanation could be that in the case of the “Zn/Cs/SBDS (1.5 mM)” sample, the system failed to reach equilibrium during the elapsed time. In the case of the “Zn/Cs/SDS (30 mM)” sample, the better corrosion resistance is also supported by the higher Rp and lower icorr values.

Changes:

The OCP-time curve of both samples are presented in the appendix in Fig. A5b.

Comment #8:

Authors are advised to fit the Nyquist plots using a suitable equivalent circuit and compare the resulting charge transfer resistance of the cases and use it to calculate the IE and add them in Table 2. You can refer to the following paper in your discussion (Phosphate glass doped with niobium and bismuth oxides as an eco-friendly corrosion protection matrix of iron steel in HCl medium…)

Answer #8:

The authors thank the Reviewer this advice. The experimental impedance data were fitted to electrical equivalent circuits. The best fit was obtained with an Rs(Ccoat(Rcoat)(QdlRct)) circuit, where Rs is the resistance of the electrolyte, Ccoat and Rcoat are attributed to the coating, while Qdl and Rct represent the charge transfer at the electric double layer. In all cases, the accuracy of the fitting was evaluated using the chi-squared (χ²) value, which was approximately 10⁻³ or less. The inhibition efficiency values were also calculated with this alternative way, and are in a good agreement with the IE values obtained by Eq. 3. These findings were added to the manuscript as well as the suggested reference. The figure containing the fitted curves has been added to the appendix as Fig. A5a.

Changes:

“To gain more information about the corrosion process, an equivalent circuit model was fitted to the Nyquist-plot of the EIS spectra using the ZSimpWin software. The accuracy of the fitting was evaluated using the chi-squared (χ²) value, which was approximately 10⁻³ or less. Using the polarization resistance (Rp) values ​​obtained from fitting provide an alternative method for calculating the inhibition efficiency (IE) with Eq. 5 [43]:

(5)

where is the polarization resistance value of the bare Zn sample, and is the polarization resistance of the coated samples.” (Line 507)

“To gain more information about the corrosion process, an equivalent circuit model was fitted to the points of the impedance spectra (the fitted curves are plotted in Fig. A5. in the Appendix). The figure of the equivalent circuit is presented in Fig. 6b, while the fitted parameters are summarized in Table 2. From the polarization resistance (Rp) values the inhibition efficiency (IE) of the coatings can be calculated with Eq. 5.” (Line 316)

Table 3 was added to Section 2.6

Fig. A5 was added to the Appendix.

Comment #9:

Also, it is advised to determine anodic and cathodic Tafel slopes from Fig. 6d and discuss the values to give insights into the corrosion determining reaction (anodic or cathodic). You can take guide from (Electropolymerized conducting polyaniline coating on nickel-aluminum bronze alloy for improved corrosion resistance in marine environment).

Answer#9:

The authors are thankful for this comment. In the case of the tested samples, the cathodic branches of the polarization curves exhibit a low slope (bc → ∞), indicating diffusion control. Consequently, only the anodic branches of the polarization curves were utilized to determine the corrosion current density. These values ​​have been added to Table 2. as well as a brief explanation to the text.

Changes:

An additional column was added to Table 1.

„The cathodic branches of the polarization curves according to the Tafel representation curves exhibit a low slope, indicating diffusion-controlled process. The slopes of the anodic branches (ba) are summarized in Table 3.” (Line 328)

Comment #10:

Typo errors e.g. L54 (and cosmetics), avoid conjunctions as (so..),

Answer #10:

The authors thank these suggestions. The typo and the conjunctions were corrected at the appropriate places.

Comments on the Quality of English Language

minor editing

Answer:

The manuscript was read through and corrected several places.

Changes:

Wording and spelling errors were corrected in lines 35, 39, 40, 46, 48, 54, 55, 56, 61 and numerous other places.

Reviewer 3 Report

Comments and Suggestions for Authors

The author should improve the following parts of manuscript.

1. AFM should additionally determine the thickness of deposited layer to evaluate the correctness of values calculated by using Eq 2

2. AFM micrograpghs are rather poor quality (Fig.4). I guess that there are layers deposited on glass (in the caption there is a lack of this info). I suggest using a supersharp tip (< 2nm) to obtain high- resolution images. Moreover, the inset box extracted from flat areas (dark colour) should provide better presentation quality. AFM images should be carried out for 5 layers as shown in Fig.3. Surface roughness (Sq) should be determined. It is unclear what it means “roughness factor is below 1.005”. Furthermore, AFM  images should be taken also for samples after immersion in water at 600 s (Fig.3)

3. Depth profiles (Fig.5) are presented incorrectly. They should be as long presented as Si signal appears and S signal decreases. Presented depth profiles do not correlate with thickness values depicted in Table 1. On the layer/Si interface, we should expect decreases in intensity of S and increases  Si.

4. The authors should provide XPS spectra. Detailed discussion of the chemical nature (chemical bonds) observed in XPS spectra is mandatory.

5. Legend in Fig.5 contains Gls/CS/SDS (10mM). I assume that it is a typo. 

6. AFM micrographs and XPS data for layers deposited on Zn are needed. AFM and XPS data should be helpful for estimation of porosity and determination of the influence of surface chemistry on corrosion properties.

Author Response

The authors are grateful for the opportunity to improve the manuscript based on the Editors and Reviewers' comments. The answers to the questions and comments and the list of changes can be found below. The comments are in bold, and the responses and list of changes is provided for each comment. The changes were marked in the manuscript and appendix in red, and the deleted parts are red and crossed out. A clean manuscript is also provided, without the deleted parts and the red marks. Whenever a line is referred to in this document, it refers to the marked version of the manuscript.

Comments and Suggestions for Authors

The author should improve the following parts of manuscript.

Comment #1:

AFM should additionally determine the thickness of deposited layer to evaluate the correctness of values calculated by using Eq 2.

Answer#1:

The authors are thankful for this suggestion. The thickness of the coatings were determined by fitting a thin layer optical model to the transmittance spectra of the coatings, described in Ref 34. These values are now included in Table 1. In order to visualize the effect of surfactant additives on the layer thickness, the relative increase of layer thickness was calculated from the obtained thickness with respect to the value of the unmodified chitosan coating (this process is described by equation 2).

During the additional AFM measurements, the layer thickness for one sample was determined using the method suggested by the reviewer (we scratched the coating, then calculated the thickness of the layer from the data of the depth profile measured perpendicular to the direction of the scratch). The layer thickness was obtained by averaging of 45 values determined over a 20×20 μm area. The obtained value of 357±1 nm is in good agreement with the layer thickness value in Table 1. The description of this process and the results were added to the manuscript and Appendix.

Changes:

„As an example, to validate the layer thickness values ​​in Table 1, in the case of a “Gls/CS+SDBS (0.1 mM)” sample, the layer thickness was also determined by AFM (based on the depth profile taken from the scratched coating, see Fig. A3 in Appendix). The obtained value (357±1 nm) is in good agreement with the layer thickness value in Table 1. (378±10 nm).” (Line 249)

„In order to validate the layer thickness values ​​obtained from optical measurements, in one case the layer thickness of a sample was also determined with AFM, in such a way that the sample was scratched and then the depth profile was recorded perpendicular to the scratch.” (Line 479)

“In order to validate the layer thickness determined with UV-Vis spectroscopy, a sample was scratched and the depth profile was recorded perpendicular to the scratch. The profile and the calculated layer thickness (average over a 20×20 μm area, which is sufficiently close to the value in Table 1.) are presented on Fig. A3.” (Line 618)

Fig A3. was added to the Appendix.

Comment #2:

AFM micrograpghs are rather poor quality (Fig.4). I guess that there are layers deposited on glass (in the caption there is a lack of this info). I suggest using a supersharp tip (< 2nm) to obtain high- resolution images. Moreover, the inset box extracted from flat areas (dark colour) should provide better presentation quality. AFM images should be carried out for 5 layers as shown in Fig.3. Surface roughness (Sq) should be determined. It is unclear what it means “roughness factor is below 1.005”. Furthermore, AFM images should be taken also for samples after immersion in water at 600 s (Fig.3)

Answer #2:

The authors thank the Reviewer this comment and suggestion. The major aim of the AFM characterization was to make it evident that the significant changes of wetting properties were not caused by changes in surface roughness or heterogeneity, but by the applied surfactants (chemical composition). From this point of view, the large-scale AFM micrographs are informative. However, we are planning to purchase and try high resolution AFM tips on these surfaces in the future (as well as the AFM measurements after immersion). The inset boxes were changed to rulers for clarity.

AFM measurements were also carried out on the two remaining samples on glass substrate (this is now included in the caption of Fig. 3 also). The AFM micrographs were added to Fig. 3. Wenzel roughness factor (the ratio between the actual and projected surface area) is used in the Wenzel wetting model: it gives the correlation between the cosine of apparent contact angles measured on a rough and on a smooth surface of the same material. The definitions of the roughness factors and the determined Sq values were added to the manuscript.

Changes:

“Nevertheless, the surface roughness of the coatings (determined in 20×20 μm areas) is not significant. The Wenzel surface roughness factors (ratios between the actual and projected surface area, used in the Wenzel wetting model [36]) are below 1.001 in all cases, while the Sq surface roughness values (quadratic mean of profile height deviations from the mean line) are (in order from a to e) 2.3 nm, 2.0 nm, 4.1 nm, 5.6 nm and 2.0 nm. This means that it is not the morphology, but the chemical composition that significantly influences the measurable value of the contact angles.” (Line 244)

Comment #3:

Depth profiles (Fig.5) are presented incorrectly. They should be as long presented as Si signal appears and S signal decreases. Presented depth profiles do not correlate with thickness values depicted in Table 1. On the layer/Si interface, we should expect decreases in intensity of S and increases Si.

Answer #3:

The authors thank the Reviewer this observation. The spectra were previously recorded until the size of the sulfur peak decreased significantly, only the last points were not included in the previous figure. This has been corrected, and the new figure shows all points as suggested by the reviewer. Knowing the new data points, the scale showing the depth was recalculated as follows. The atomic percentage calculated from the spectra were assigned to the number of etching cycles, then the layer thickness values ​​“from Table 1.) were divided by the number of etching cycles required to completely remove the coating. Assuming that the same thickness of coating layer was removed during each cycle, the values ​​of the depth of recording each spectra can be determined.

Changes:

 „The depth values ​​were calculated by dividing the layer thickness values ​​in Table 1. by the number of etching cycles required to completely remove the coating (it was assumed that the same thickness of coating layer was removed during each etching cycle).” (Line 266)

“Reaching the glass substrate was identified by the appearance of the silicon peak in the XPS spectra and the decrease of the sulfur peak close to zero.” (Line 263)

Figure 5. was revised and expanded with the additional data points.

Comment #4:

The authors should provide XPS spectra. Detailed discussion of the chemical nature (chemical bonds) observed in XPS spectra is mandatory.

Answer #4:

The authors are thankful for this suggestion. The aim of XPS was primarily to follow the depth distribution of the surfactant in the layer by monitoring the sulfur content. For this, it was sufficient to calculate the atomic percentage of sulfur at each measurement depth from the integral of the peaks of the elements found in the layer, and the decomposition of the peaks was not necessary. However, the analysis of the peaks provides useful and essential information, so the decomposition of a carbon and sulfur peak obtained from an example spectrum is included in the Appendix, together with the details and interpretation of the types of chemical bonds obtained from it.

Changes:

„Examples for the decomposition of the carbon and sulfur peaks in the case of a “Gls/CS+SDS (0.5 mM)” sample are presented in Fig. A4. The peak fitting was performed in CasaXPS software. The binding energy was set by fixing the C–(C,H) component of the C 1s peak at 284.8 eV. The carbon high resolution spectra were decomposed by Lorentzian Asymmetric Lineshape (LA) to the following 4 components: at 284.8 eV, typical of carbon only bound to carbon and hydrogen; near 286.3 eV, typical of carbon making a single bond with oxygen or nitrogen; near 287.8 eV typical of acetal and amide; near 288.8 eV typical of carboxyl group (all of the above are characteristic of chitosan) [46,47]. The sulfur spectra were decomposed by GL(30) peak shape (Gaussian-Lorentzian product function with a mixing parameter of 30). The splitting value for 2p3/2 and 2p1/2 was 1.2 eV. The area ratio between the 2p 3/2 and 2p 1/2 peaks was set to be 2:1. The binding energy belonging to 169.8 eV and 168.6 eV can be attributed to SO4 species presented in the surfactant [48,49].” (Line 627)

Comment #5:

Legend in Fig.5 contains Gls/CS/SDS (10mM). I assume that it is a typo.

Answer #5:

The authors are thankful for the observation. The Reviewer was right, it was a typo that has been corrected.

Changes:

The legend of Fig.5 was modified.

Comment #6:

AFM micrographs and XPS data for layers deposited on Zn are needed. AFM and XPS data should be helpful for estimation of porosity and determination of the influence of surface chemistry on corrosion properties.

Answer #6:

The authors thank the Reviewer this suggestion. As mentioned above, the primary aim of AFM measurements was to study the possible effect of morphology on wettability. In addition, the reason for the use of zinc substrate is also a measurement technical reason: electrochemical measurements can only be performed in the case of coatings applied to conductive substrates. Although, the measurements suggested by the Reviewer can reveal useful information about the behaviour of the samples during a possible application, however, it was not possible to perform them before the specified deadline, but they are included in our future plans.

Round 2

Reviewer 2 Report

Comments and Suggestions for Authors

Authors have addressed my concerns. I suggest publication

Reviewer 3 Report

Comments and Suggestions for Authors

The authors corrected the manuscript in an accepted way. I recommend this paper for publication.

Comments on the Quality of English Language

Minor editing of English language required